https://doi.org/10.1038/s41467-021-21928-4　　**OPEN**

# Circulating mucosal-associated invariant T cells identify patients responding to anti-PD-1 therapy

Sara De Biasi [1,9 ✉], Lara Gibellini [1,9], Domenico Lo Tartaro [1,9], Simone Puccio [2], Claudio Rabacchi[1], Emilia M. C. Mazza [2], Jolanda Brummelman[2], Brandon Williams[3], Kelly Kaihara[3], Mattia Forcato[4], Silvio Bicciato [4], Marcello Pinti[4], Roberta Depenni[5], Roberto Sabbatini[5], Caterina Longo[6], Massimo Dominici [1,5], Giovanni Pellacani [1], Enrico Lugli [2,7,10] & Andrea Cossarizza [1,8,10]

Immune checkpoint inhibitors are used for treating patients with metastatic melanoma. Since the response to treatment is variable, biomarkers are urgently needed to identify patients who may benefit from such therapy. Here, we combine single-cell RNA-sequencing and multiparameter flow cytometry to assess changes in circulating CD8$^+$ T cells in 28 patients with metastatic melanoma starting anti-PD-1 therapy, followed for 6 months: 17 responded to therapy, whilst 11 did not. Proportions of activated and proliferating CD8$^+$ T cells and of mucosal-associated invariant T (MAIT) cells are significantly higher in responders, prior to and throughout therapy duration. MAIT cells from responders express higher level of CXCR4 and produce more granzyme B. In silico analysis support MAIT presence in the tumor microenvironment. Finally, patients with >1.7% of MAIT among peripheral CD8$^+$ population show a better response to treatment. Our results thus suggest that MAIT cells may be considered a biomarker for patients responding to anti-PD-1 therapy.

[1] Department of Medical and Surgical Sciences for Children & Adults, University of Modena and Reggio Emilia, Modena, Italy. [2] Laboratory of Translational Immunology, IRCCS Humanitas Research Hospital, Milan, Italy. [3] Bio-Rad Laboratories, Hercules, CA, USA. [4] Department of Life Sciences, University of Modena and Reggio Emilia, Modena, Italy. [5] Department of Oncology, University of Modena & Reggio Emilia, Modena, Italy. [6] Department of Surgery, Medicine, Dentistry and Morphological Sciences, University of Modena and Reggio Emilia, Modena, Italy. [7] Humanitas Flow Cytometry Core, IRCCS Humanitas Research Hospital, Milan, Italy. [8] National Institute for Cardiovascular Research, Bologna, Italy. [9]These authors contributed equally: Sara De Biasi, Lara Gibellini, Domenico Lo Tartaro. [10]These authors jointly supervised this work: Enrico Lugli, Andrea Cossarizza. ✉email: debiasisara@yahoo.it

CD8[+] T cells can drive adaptive immune responses against several types of human malignancies, in particular those with higher mutational burden and neoantigen load[1]. These cells are activated by tumoral antigens, undergo expansion, and can localize and kill infected or cancer cells. However, prolonged exposure to cognate antigens often contracts the effector capacity of T cells and attenuates their therapeutic potential. This process, collectively known as T-cell exhaustion, is characterized by limited proliferation, cytokine production and effector capacity, metabolic rearrangement, increased inhibitory receptors expression and genome-wide accumulation of epigenetic modifications at effector and memory-related gene loci[2]. Among inhibitory receptors, programmed death-1 (PD-1) has been extensively studied, and is now targeted by therapies with monoclonal antibodies that are capable to reinvigorate T cells in several cancer settings. However, immune checkpoint inhibitors (ICI) mediate tumor regression only in a subset of patients, and the mechanisms at the basis of therapeutic resistance are poorly known[3]. A number of studies have initially focused on the mutational load of the tumor as well as on quality of the cells infiltrating the tumor microenvironment, and revealed that increased mutational burden and the presence of CD8[+] T cells with stem-like qualities[4,5], among others, can predict the response to ICI[6–10]. However, tumoral tissue may not be always accessible, thereby making the quest of circulating biomarkers an absolute need. In this regard, recent studies have shown that responding patients have larger clones (those occupying >0.5% of repertoire) post-treatment than non-responding patients or controls, and this correlates with effector memory T-cell percentage[11], suggesting that peripheral T-cell expansion could predict tumor infiltration and clinical response[12].

Over the last decade, a pressing need to deeply interrogate immune cells either in the tumor microenvironment and/or in blood has led investigators to integrate data obtained from traditional approaches with those obtained with new, more advanced, single-cell technologies, capable to define characteristics of immune cells at an unprecedented degree of resolution[13].

Using single-cell RNA sequencing (scRNA-seq) and high-dimensional flow cytometry, here we show that mucosal-associated invariant T (MAIT) cells are increased in patients with metastatic melanoma who respond to anti-PD-1 therapy. The identification of MAIT cells as biomarkers for patients responding to such treatment could represent an useful tool to tumor immunotherapy and to maximize patient's benefit from this treatment.

## Results

**A higher proportion of activated effector memory CD8[+] T cells in responders.** We initially used high-dimensional flow cytometry to longitudinally define the characteristics of T cells upon PD-1 blockade in melanoma patients (Supplementary Fig. 1). Computational analysis of aggregated data from multiple patients and time points identified 28 clusters (individually labeled as C) among CD8[+] T cells, resolving a broad spectrum of T-cell states, including maturation, activation and exhaustion. C21, C22, C26 display phenotypic identity proper of subsets of naive T cells, characterized by expression of CD45RA, CCR7, CD27, CD28, negligible expression of CD25 and ICOS, and absence of additional markers[14] (Fig. 1A). C28 represents recently activated T cells characterized by expression of CD38 and ICOS, but no expression of the late activation marker HLA-DR. T stem-cell memory cells are identified in C10, and their phenotype is similar to that of naive T cells, and includes the expression of CD95[5,15]. C1, C20, and C27 represent CCR7[+]CD45RA[−] central memory T cells characterized by high expression of CD28, CD27, BTLA, CD194, CD25, CD95 and ICOS. C1 displays high levels of

CD194, CD28, CD95 while C20 represents a cluster of T central memory (TCM) cells that expresses high level of CD39. C6, C17, C11, C2, C24, and C5 represent terminally differentiated T cells, being characterized by the expression of CD45RA, but not of CCR7, CD27, or CD28, and high levels of CD244, CD57 and T-bet. These cells also lacked granulysin expression.

C15, C4, C25, C14, C9, C13, C12, C19, and C16 represent effector memory T-cell subsets characterized by the lack of expression of CD45RA, CCR7, and expression of CD25 and CD95[16]. Among these, C14 expressed PD-1 and CD57, and T-bet at intermediate levels, thereby suggesting the identification of replicative senescent cells[17]. C9, C13, C12 are transitional effector memory T cells as they express intermediate levels of CD28 and CD27[18]. C9 expresses CXCR6, identifying effector memory cells with the capability to migrate to metastasis[19], while C12 is a cytotoxic T-cell subset displaying high level of granulysin. C19 display high levels of CD127, CD39, and CD25, identifying not only metabolically activated, but also tumor-reactive cells[20]. C16 is a cluster of activated and proliferating effector memory T cells characterized by high level of expression of Ki67, ICOS, CD95, HLA-DR, CD71, CD98, CXCR6, granulysin, CD38, intermediate expression of CD127, CD39, CD25, CD28, CD194, CD27, BTLA, T-bet and CD244; as shown in Fig. 1B, this cluster was much more represented in responder patients if compared to non-responders.

Longitudinal analysis did not identify obvious differences in the dynamics of these immune populations between responders and non-responders to anti-PD-1 therapy (Supplementary Fig. 2). Cross-sectional analysis identified C16, highly proliferating Ki67[+]CD71[+] effector cells equipped for cytotoxicity (GNLY[+]), whose relative proportion was higher in responder before starting therapy ($p < 0.001$) (Fig. 1B). This difference remained stable also after treatment ($p < 0.01$) (Fig. 1B).

**MAIT cells are more abundant in responders as revealed by scRNA-seq.** To further define the dynamics of T cells potentially involved in therapeutic response, we performed scRNA-seq of isolated CD3[+]CD8[+] T cells from a total of 20 patients at T0, T1, and T2 after anti-PD-1. After quality control (see Methods), 56,142 cells were deemed suitable for analysis. Contaminating 4210 NK and 231 monocytes cells, expressing TYROBP, FCGR3A, KLRB1, and LYZ, respectively, were removed from the analysis. We obtained a total of 51,701 purified CD8[+] T cells. Using a cTP-net, a deep neural network trained on multi-omics data, we imputed surface protein abundances within the scRNA-seq data to confirm T-cell phenotype[21] (Supplementary Fig. 3).

Computational analysis identified eight different cell clusters on the basis of gene expression profiles (Fig. 2A, B; see also Source File). Naive T cells were identified by expression of LEF1, SELL, TCF7 genes while terminally differentiated effector memory cells, with cytotoxic properties were characterized by the expression of GZMB, GNLY, NKG7, EFHD2, and CXCR3[22]. Two different clusters of effector memory cells were recognizable: one cluster of transitional effector memory (characterized by the expression of GZMK and LYAR) and one of more mature and activated phenotype with homing properties (expression of TNFAIP3, CXCR4, CREM, CD69)[23]. Two clusters of recently activated naive T cells have been characterized: one expressed GATA3 and IL7R, the other FOS and JUN. Activated and replicating effector memory T-cell clusters were identified by the expression of HLADRA, HLADRB1, CD74, GZMA, PCNA, MKI67, TOP2A, MCM4, MCM. Finally, mucosal-associated invariant T (MAIT) cells with homing properties were identified they expressed high level of KLRB1, SLC4A10, MAF, and CXCR4[10,24].

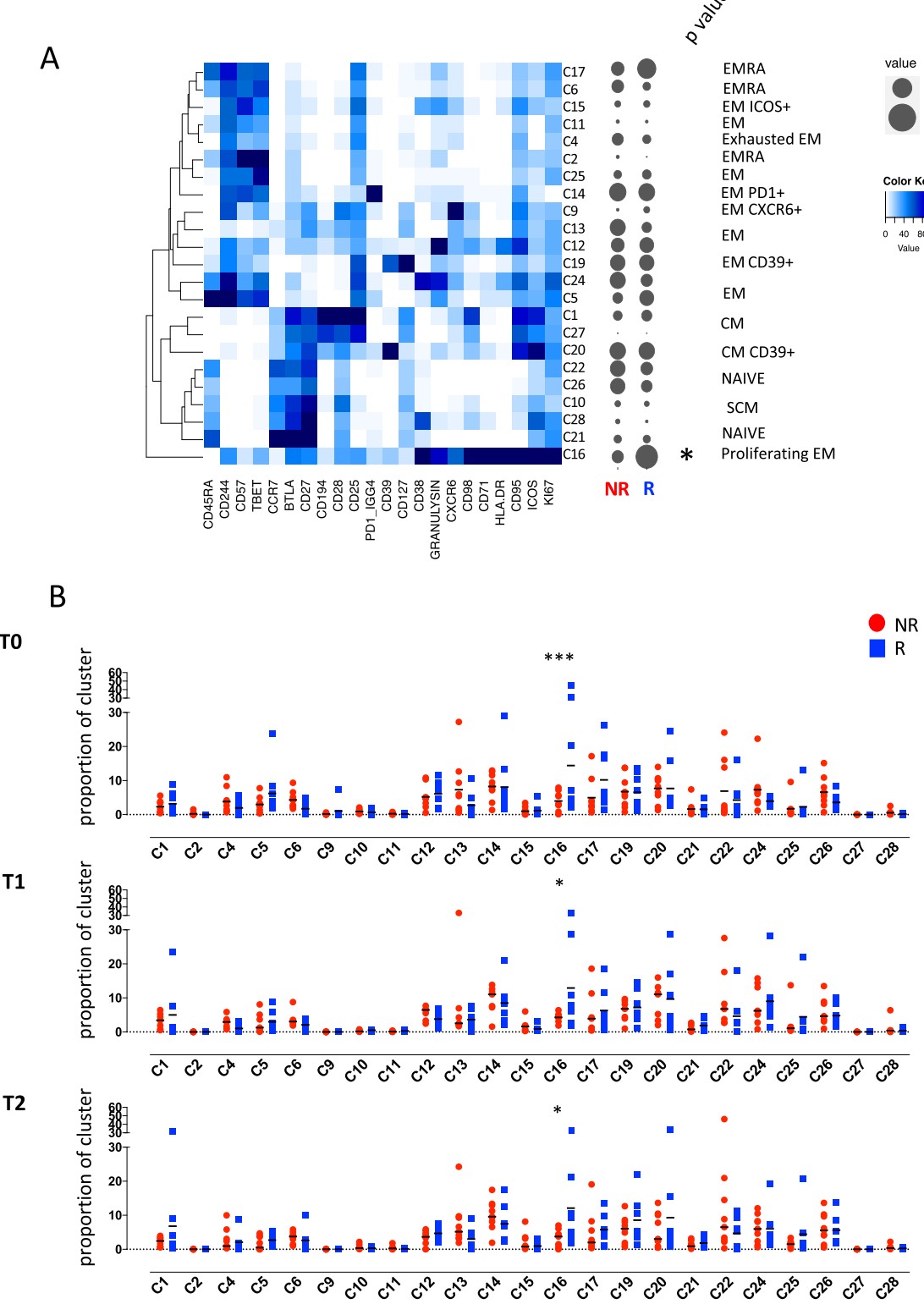

**Fig. 1 High-dimensional single-cell analysis of CD8$^+$ T cells identifies higher proportion of activated effector memory T cells in responders. A** Heatmap showing the iMFI of specific markers in discrete Phenograph clusters. Ballons indicate the median frequency of each cluster amongst responders and non-responders. Statistical analysis by two-sided Mann–Whitney nonparametric test, Bonferroni's multiple comparisons test, *$p = 0.046$. **B** Proportion of cell within each cluster for each individual. Data represent individual values, mean (center bar) ± SEM (upper and lower bars). Statistical analysis by two-sided Mann–Whitney nonparametric test, Bonferroni's multiple comparisons test; T0, ***$p < 0.0001$; T1, *$p = 0.046$; T2 $p = 0.037$. T0 = before therapy, T1 = after 1 cycle of therapy, T2 = after two cycles of therapy. Individual measurements of NR = 9, R = 8. Source data are provided as a Source Data file.

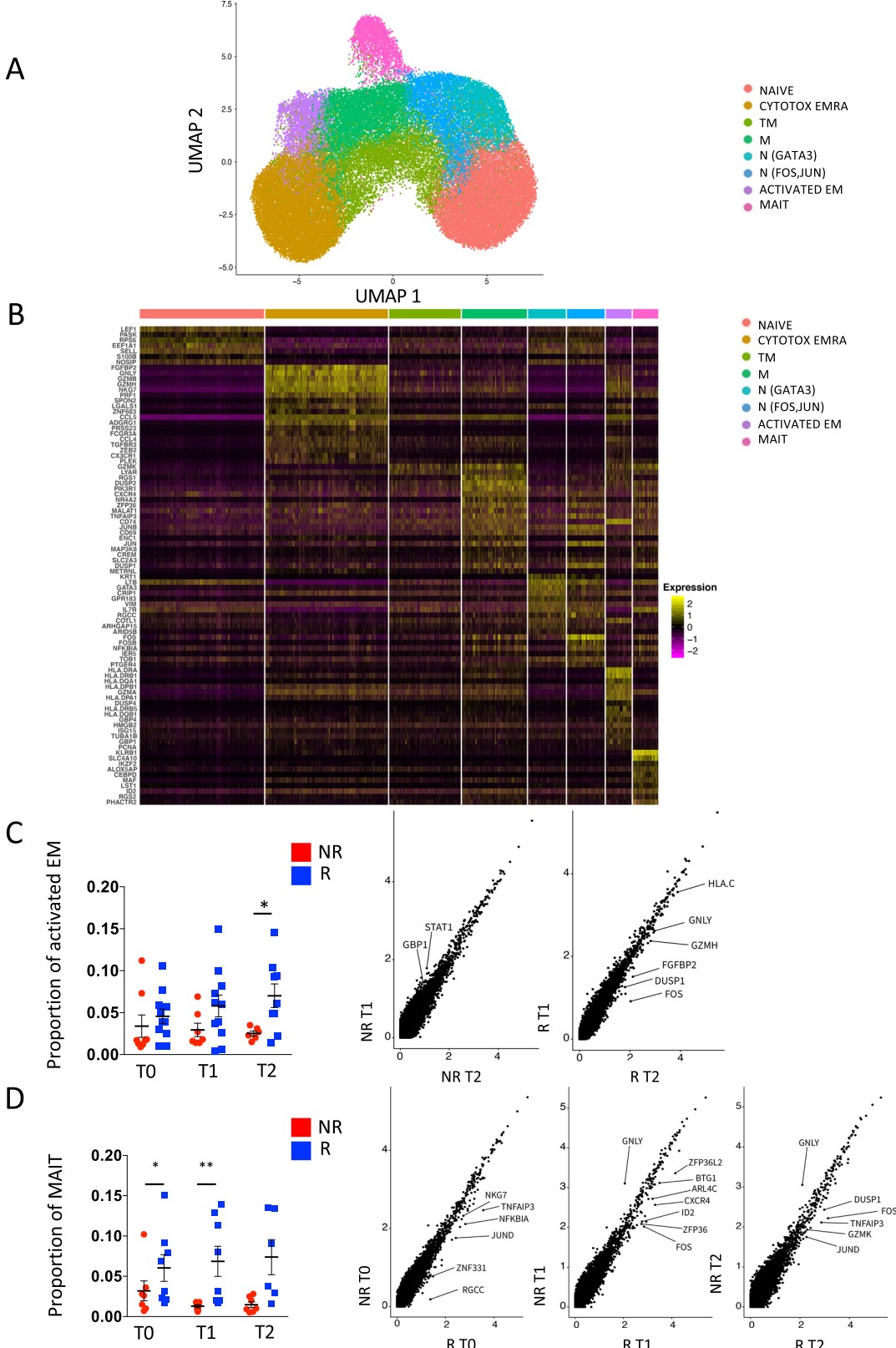

Pseudotime analysis revealed that the differentiation process started from naive T cells towards terminally differentiated T cells passing through activated naive T cells, transitional effector memory T cells and effector memory T cells (Supplementary Fig. 4). In this process, the transcriptionally distinct MAIT cells belong to a different branch of the Pseudotime trajectory compared to the rest of the T cells, albeit mapping close to effector memory T cells, in line with their shared phenotypic identity[25].

No main differences were found between R and NR in the proportion of naive, cytotoxic terminally differentiated and activated naive T cells, both before and after therapy, as revealed

**Fig. 2 MAIT cells are more abundant in responders as revealed by scRNA-seq. A** UMAP plot depicting CD8$^+$ T-cell heterogeneity. Cells are colored according to the eight clusters defined in an unsupervised manner. **B** Heatmap displaying scaled-expression values of discriminative gene set per cluster related to CD3$^+$CD8$^+$ T cells that passed quality control. A list of the most representative genes is shown per each cluster (left). N, naive; EMRA, effector memory expressing CD45RA; TM, transitional memory; M, memory; EM, effector memory; MAIT, mucosal-associated invariant T cells. **C** Proportion of activated effector memory (EM) CD8$^+$ T cells at different time points (left) and differential gene expression in this cluster between responders and non-responders at T2 (right). *p*-values of the differential expression analyses are reported in source tables. Only genes differentially expressed are reported in the figure. Statistical analysis by Mann–Whitney nonparametric test, Bonferroni's multiple comparisons test; *$p = 0.046$. **D** Proportion of MAIT cells and differential gene expression of this cluster between responders and non-responders at T0, T1, and T2 (right). *p*-values of the differential expression analyses are reported in source tables. Only genes differentially expressed are reported in the figure. Data represent individual values, mean (center bar) ± SEM (upper and lower bars). Statistical analysis by two-sided Mann–Whitney nonparametric test, Bonferroni's multiple comparisons test; if not indicated, *p*-value is not significant. *$p = 0.023$; **$p = 0.0012$. Source data are provided as a Source Data file. T0 = before therapy, T1 = after 1 cycle of therapy, T2 = after two cycles of therapy. Individual measurements of NR = 8, R = 11.

by analysis of gene expression profiles by scRNA-seq (Supplementary Fig. 5). The proportion of activated effector memory T cells, reminiscent of C16 as defined by flow cytometry, was higher after two cycles of therapy in R compared to NR (Fig. 2C, left panel). At the same time, activated effector memory T cells from R expressed higher levels of genes indicating activation (*FOS, DUSP1, FGFBP2, HLAC*) and cytotoxic behavior (*GNLY, GZMH*), thereby suggesting heightened functional capacity in R (Fig. 2C, right panels).

The proportion of MAIT cells was higher in R before therapy and after the first cycle of therapy (Fig. 2D, left panel). This trend was visible also after the second cycle of therapy. Similarly to EM T cells, also MAIT cells showed overexpression of genes related to cell activation in R compared to NR before (*TNFAIP3, NKG7, NFKBIA, JUND, ZNF331, RGCC*) or after the first (*ZFP36L2, BTG1, ARL4C, CXCR4, ID2, FOS, ZFP36*) or the second cycle (*DUSP1, FOS, TNFAIP3, GZMK, JUND*) of anti-PD-1 therapy (Fig. 2D, right panels), overall suggesting a dynamic regulation of MAIT cell activation over time.

**Activated MAIT cells with homing properties are more abundant in responders.** We further subclustered MAIT cells before and after therapy to gain more insights into the cellular dynamics of these cells during the anti-tumor immune response. Our approach identified two different types of MAIT cells with differential expression of genes related to T-cell activation or effector functions *DUSP1, ZFP36, TNFAIP3, ZFP36L2, FOS, CXCR4, NFKBIA, CD69, TSC22D3, BHLHE40* and *JUN*, thereby suggesting the identification of quiescent and activated subsets of cells (Fig. 3A, B). In line with previous data, R showed a significantly higher proportion of activated MAIT compared to NR not only before therapy, but also after the first and the second cycle (Fig. 3C).

We next used polychromatic flow cytometry to confirm these findings also at the protein level. In this regard, we analyzed the proportion and phenotype of MAIT cells, identified as CD3$^+$CD8$^+$ T cells that expressed TCRα7.2 and CD161 (Fig. 4A, left), and found marked expansion of these cells in the circulation of R patients when compared to NR before therapy (Fig. 4A, right). This difference waned after therapy introduction in line with scRNA-seq data.

Moreover, we found that the proportion of MAIT cells expressing the homing receptor CXCR4 increased after two cycles of therapy in R, but not in NR, that had a relevant variability (Supplementary Fig. 6).

To confirm the presence of MAIT cells in the metastasis and primary tumor site we analyzed a public dataset available on Gene Expression Omnibus (GSE148190)[26]. This dataset contains single-cell RNA and TCR sequencing of PBMCs and tumor-infiltrating lymphocytes from untreated patients with metastatic melanoma. We used the scRNAseq data of blood (B),

lymph nodes metastasis (LN), and Tumor (T) from patients K383, K409, and K411. A total of 26,757 cells have been analyzed (11,614 of B, 12,915 of LN, and 2170 of T). About 3% of cells in LN and T were identified as MAIT cells expressing *CXCR4* gene, suggesting their ability to home the inflamed tissue (Fig. 4B and Supplementary Fig. 7, top). In addition, by analyzing a public dataset of CD8 T cells obtained by melanoma patients treated with ICI (see "Methods")[10], we found that MAIT cells increased in the metastatic lesions regressing after ICI compared to those that did not regress compared to baseline, thereby suggesting the potential recruitment of CXCR4-expressing MAIT cells expanding in the circulation (Supplementary Fig. 7, bottom).

We next analyzed the effector functional capacity of the MAIT cells following in vitro stimulation with IL-12, IL-18, CD3/CD28 followed by the detection of the effector molecules GRZM-B, IFN-γ, and TNF (Supplementary Fig. 8). The overall quality of the response of MAIT cells, as assessed by combinatorial cytokine production, was largely similar between R and NR at different time points, where the majority of cells were capable to simultaneously produce GRZM-B, IFN-γ, and TNF. Moreover, R were characterized by higher proportion of cells producing IFN-γ and GRZM-B if compared to NR (Fig. 4C). Nevertheless, before therapy, the proportion of cells able to produce only GRZM-B was higher in R if compared to NR (Fig. 4D), thereby corroborating previous evidence that MAIT cells show preferential effector propensity.

**Level of MAIT cells before therapy identifies responder patients.** We next evaluated the prognostic significance of the levels of MAIT cells in the circulation as predictive biomarker of the response to anti-PD-1 therapy. Flow cytometric analysis revealed that, within CD8$^+$ T cells, the median level of MAIT in the population of patients with metastatic melanoma was 1.7%, thus this value was used as a cutoff to stratify patients. Figure 5 reports that patients with a frequency of MAIT cells >1.7% had an increased probability to respond than those patient with MAIT cells <1.7% ($p = 0.0363$, Log-rank Mantel-Cox test).

## Discussion
The main finding of our study is that patients who respond to ICI are characterized by a different composition of T-cell subpopulations compared to those who do not respond, that are detectable before therapy initiation. The most relevant of these differences is at the level of MAIT cells, an innate population of CD3$^+$ T cells previously involved in early immunity against infections in peripheral tissue. Although the direct role of MAIT in mediating anti-tumor immune responses in melanoma is still under scrutiny, our data suggest that investigating MAIT cell frequency in the peripheral blood could be considered a possible

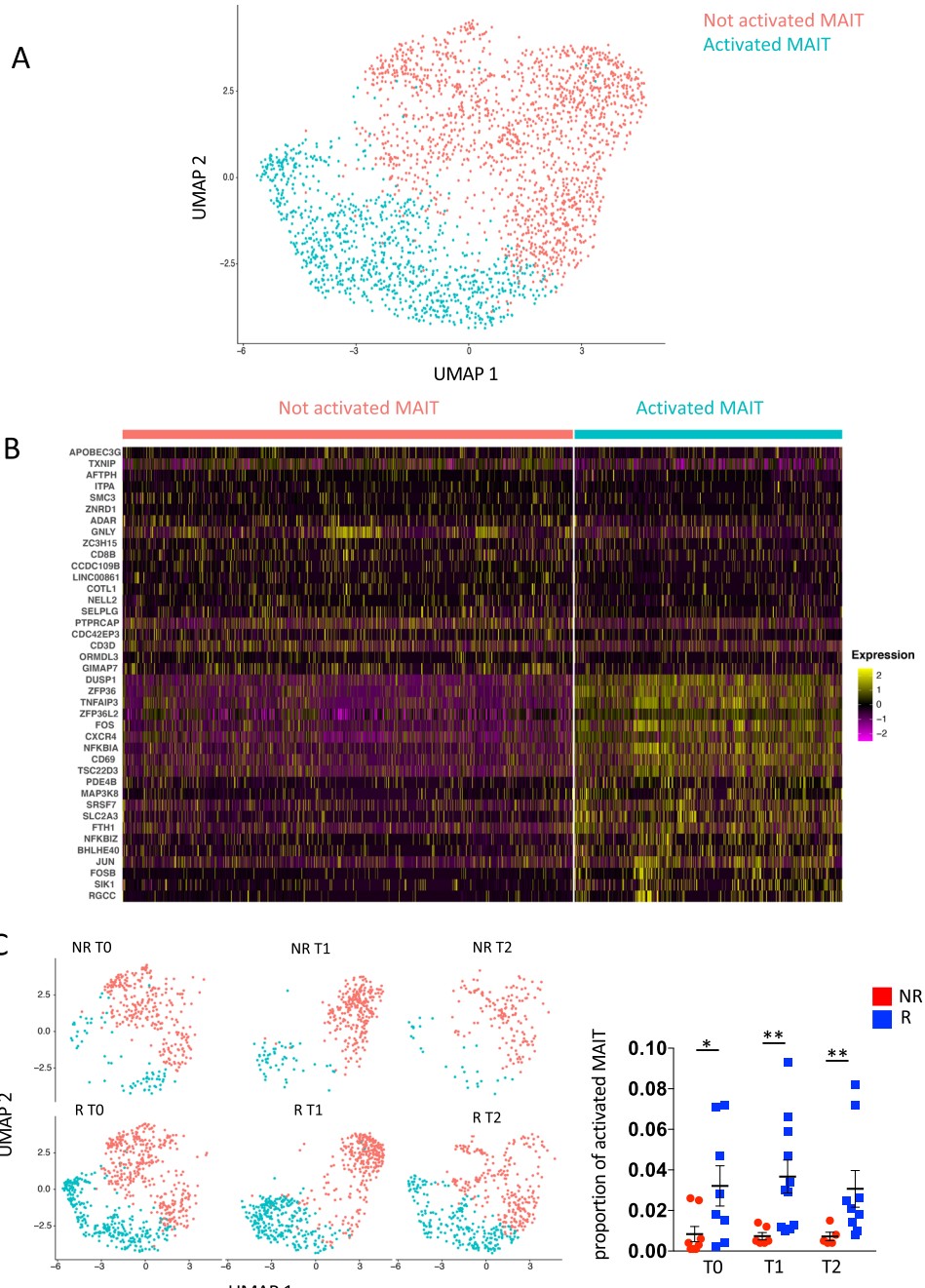

**Fig. 3 Activated MAIT cells with homing properties are more abundant in responders. A** UMAP plot of MAIT cells. Not activated MAIT are in salmon and activated ones are in light blue. **B** Heatmap displaying scaled-expression values of discriminative gene set per each cluster of MAIT cells. A list of representative genes is shown on the left. **C** Left part: UMAP plot representing two clusters of MAIT cells between R and NR at T0, T1, T2. Right part: Proportion of activated MAIT cells between R and NR at T0, T1, T2. *$p = 0.04$; T1, **$p = 0.005$; T2, **$p = 0.007$. Statistical analysis by two-sided Mann–Whitney nonparametric test, Bonferroni's multiple comparisons test. T0 = before therapy, T1 = after 1 cycle of therapy, T2 = after two cycles of therapy. Individual measurements of NR = 8, R = 11. Source data are provided as a Source Data file.

predictive marker of successful therapy. Following introduction of ICI, R show different proportions of T cells compared to NR, involving the expansion of activated effector memory cells showing features of immune activation, proliferation and effector differentiation, as previously reported by other groups[27].

During the last decade, the immune response mediated by T cells in cancer patients assuming ICI has been deeply investigated by analyzing both tumor-infiltrating lymphocytes and circulating T cells. Patients with melanoma or non-small cell lung cancer are characterized by an exhausted T-cell phenotype along with impaired proliferation and low metabolic activation, and a high oligoclonal repertoire[28–30]. Activation of CD8[+] T cells has been considered a hallmark of response to therapy, and indeed after one cycle of therapy, Ki67 (a marker of cell proliferation) increases among effector memory cells[27,31].

We show here that even if before treatment R and NR were characterized by similar clinical characteristic in terms of tumor burden and LDH level, activated effector memory T cells were more abundant in R, which can reflect a more activated CD8[+] T-cell compartment. This was particularly evident in MAIT cells.

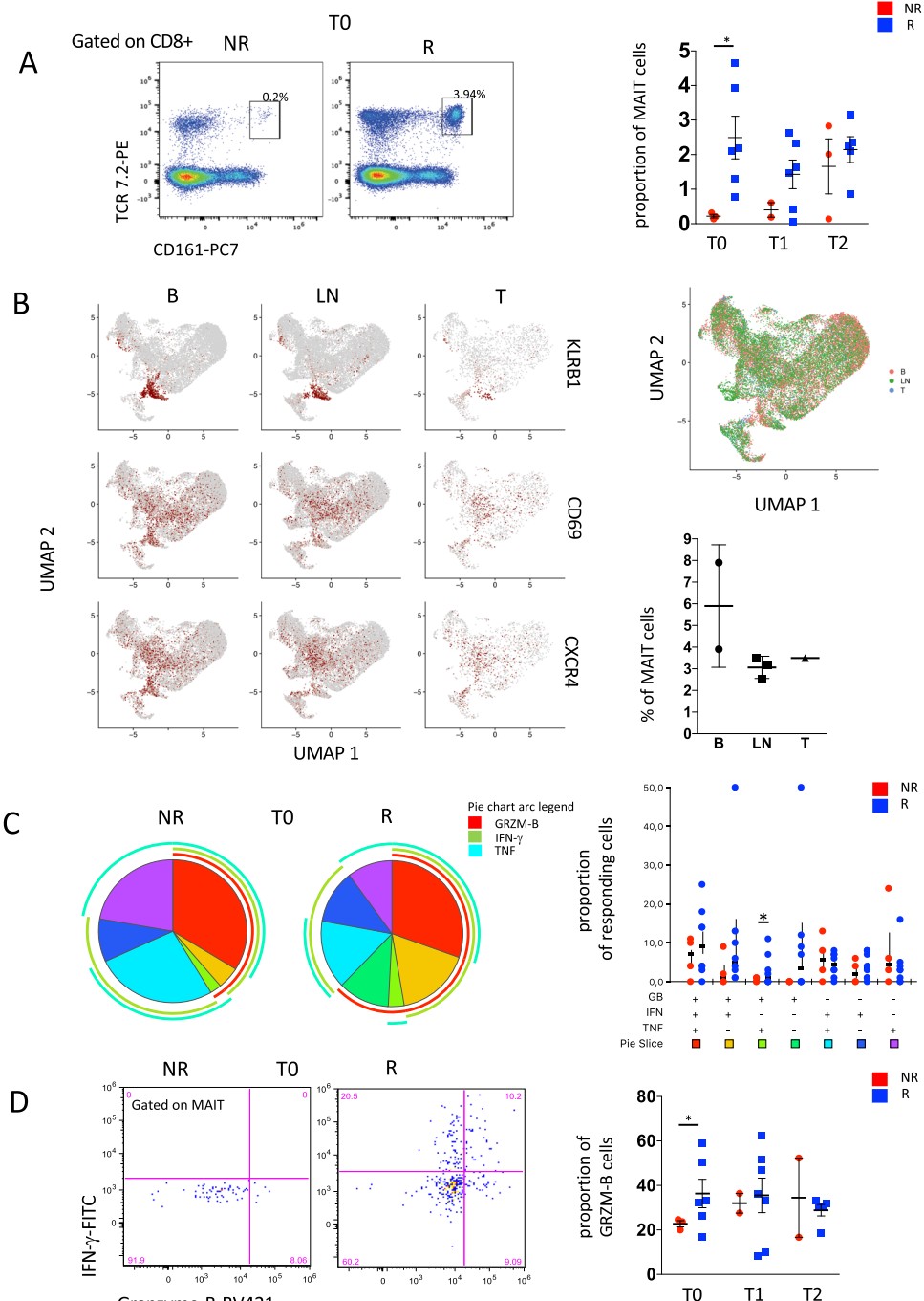

**Fig. 4 MAIT polyfunctionality evaluated after in vitro stimulation in PBMC of melanoma patients. A** Left and center panels: representative dot plots of MAIT cells, identified as TCR 7.2+ and CD161+ within CD8+ T cells of one R and one NR at T0. Right part: proportion of MAIT cells in R and NR at T0, T1, T2. Data represent individual values, mean (center bar) ± SEM (upper and lower bars). Statistical analysis by two-sided Mann–Whitney nonparametric test, Bonferroni's multiple comparisons test. *p = 0.016. NR = 4, R = 8. **B** Left panel: UMAP representation of PBMC or tumor-infiltrating lymphocytes from patients with metastatic melanoma. Expression of KLRB1, CD69, and CXCR4 MAIT cells in blood (B), lymph nodes metastasis (LN), and Tumor (T) from the K383, K409, and K411 patients (Gene Expression Omnibus, GSE148190). Right panel: UMAP representation with the distribution of cells of B, LN and T, and proportion of MAIT in blood (B), lymphnode (LN) and tumor (T). **C** Left and central panels: pie charts representing the proportion of MAIT cells producing different combinations of GRZM-B, IFNγ, and TNF after stimulation at T0. Frequencies were corrected by background subtraction as determined in unstimulated controls; permutation tests (10,000 number of permutations), using SPICE software, show no difference between R and NR. Right panel: frequency of MAIT cells expressing and producing different combinations of GRZM-B, IFN-γ, and TNF after stimulation at T0. Statistical analysis by Wilcoxon rank test; *p = 0.041. In the figure, error bars (median value) and upper whiskers (whisker range, SEM) are represented. Individual measurements of NR = 4, R = 6. **D** Left and central panels: dot plots show the difference between a R and a NR in the proportion of cells that produce IFN-γ and GRZM-B at T0. Right part, proportion of MAIT cells producing GRZM-B at different time points. Data represent individual values, mean (center bar) ± SEM (upper and lower bars). *p = 0.028, Statistical analysis by Mann–Whitney nonparametric test, Bonferroni's multiple comparisons test. Individual measurements of NR = 4, R = 6. Source data are provided as a Source Data file.

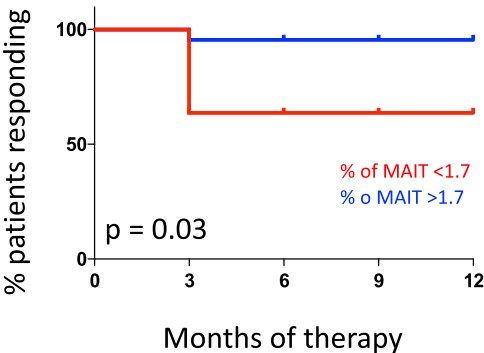

**Fig. 5 Level of MAIT cells before therapy can predict initial response to therapy.** Analysis of the cohort patients with metastatic melanoma indicates that patients with MAIT cells >1.7% of CD3+CD8+ T lymphocytes showed a better response to therapy compared to those with MAIT < 1.7% (p = 0.0363, Log-rank Mantel-Cox test). N < 1.7% = 4; N > 1.7% = 8.

Circulating MAIT cells are a pro-inflammatory and cytotoxic population within effector memory T cells[32] and can represent up to 10% of peripheral CD8+ T cells. They recognize microbial proteins presented by non-polymorphic MHC class I related-molecule (MR1) and display homing properties, as they express different homing and cytokine receptors. Furthermore, MAIT cells are deeply involved in patrolling mucosae and orchestrating the immune response in this environment[33].

The role of MAIT cells in cancer has been widely analyzed. However, few studies have investigated their role during therapy with ICI. It was found that MAIT cells were decreased in blood and displayed an altered cytokine production in patients with cervical, colorectal, gastric, hepatocellular carcinoma, lung cancer, and multiple myeloma. Moreover, controversial data exist on the prognostic benefit of MAIT cells in the tumor microenvironment, as in the case of hepatocellular carcinoma[33]. Recent studies also show that MAIT cells promote tumor initiation, growth and metastases via tumor MR1[34].

To the best of our knowledge, these are the first data that characterize MAIT cells in the peripheral blood of patients treated with anti-PD-1. We found that in R compared to NR, at baseline and after therapy introduction, (i) the proportion of MAIT cells was higher; (ii) MAIT cells displayed enhanced expression of genes related to immune activation and effector functions; (iii) the proportion of MAIT cells expressing CXCR4 was higher in R after two cycles of therapy.

CXCR4-CXCL12 axis plays an important role in the interactions between cancer cells and their microenvironment. This axis modulates the traffic of tumor cells to metastasis, and mediates invasiveness, vasculogenesis, and angiogenesis. However, preclinical melanoma models reported that this pathway can be influenced by anti-cancer treatments[35]. Hence, it is possible to hypothesize that, among other activities, the increased expression of CXCR4 on MAIT cells induced by anti-PD-1 therapy could facilitate their migration towards metastases, where they could exert a pro-inflammatory and cytotoxic activity. In line with this hypothesis, we observed that MAIT cells from R expressed CD69, which is not only an activation marker, but also a constitutively expressed marker of tissue residency. In immunotherapy-naive melanoma patients, the intratumoral presence of CD8+CD103+ CD69+ T cells that are able to significantly increase during anti-PD-1 therapy has been associated with improved survival[36]. Accordingly, we have found an increased relative proportion of MAIT cells in melanoma lesions responding to ICI compared to those that did not respond, possibly suggesting the migration of circulating MAIT cells to the tumor site, as it has been observed for conventional CD8+ T cells[27]. Future studies investigating the clonal composition of MAIT cells from the blood and the tumor will be required to confirm this hypothesis.

Recently, a rare population of MR1-restricted T cells belonging to the family of MAIT cells (defined "MR1" T cells) has been described as a rare population capable to respond to a variety of tumor cells of different tissue origin, but not to microbial antigens[37]. Thanks to its ability to kill several cancer cell lines expressing low levels of MR1 while remaining inert towards noncancerous cells, this population represents a subset with a great potential for cell therapy approaches in several malignancies[38,39].

While we acknowledge some limitations associated with this study, such as the relatively low number of patients enrolled in the study and the lack of a possible mechanism responsible of a better prognosis, nevertheless we provide evidence of the association between the frequency and the effector functions of MAIT cells and the response to ICI in melanoma, thereby suggesting that the circulating levels of MAIT cells in the peripheral blood could serve as a useful, non-invasive biomarker. Thus, further studies will be needed not only to confirm the utility of MAIT as biomarkers, but also to demonstrate their therapeutic potential or to provide actionable information about tumor's biology, which together holds great promise with respect to realizing "personalized" treatment of melanoma. In conclusion, we provide evidence of the association between the frequency and the effector functions of MAIT cells and the response to ICI in melanoma, thereby suggesting that the circulating levels of MAIT cells in the peripheral blood could serve as a useful, non-invasive biomarker. Future studies are also needed to assess whether MAIT cells are directly involved in mediating tumor regression that can be further amplified by targeting PD-1 or alternate immune checkpoints.

## Methods

**Patients.** The study was conducted on 28 patients with metastatic melanoma treated with standard-of-care nivolumab or pembrolizumab. According to the RECIST, responders (n = 17) were defined as patients with complete response (CR), partial response (PR), stable disease (SD), or mixed response (MR) of >6 months with no progression, and non-responders (n = 11) as patients with progressive disease (PD). In particular, among responders, 41.2% had CR, 35.3% had a PR, 17.6% had SD, and 5.9% (which corresponds to one patient) had a MR. The clinicopathologic characteristics of patients are reported in Table 1. The mean age of the total cohort was 71 ± 12 years and plasma lactate dehydrogenase (LDH) level was 418.7 ± 134.7. No patient had previously received other therapies.

**Table 1 Clinical characteristics of patients.**

| Variable | Non-responder (NR) (N = 11) | Responder (R) (N = 17) | Total (N = 28) |
|---|---|---|---|
| Mean age (year) | 71.0 ± 12.3 | 70.0 ± 12.4 | 71.0 ± 12.1 |
| Sex (%M) | 45.4 | 76.4 | 64.3 |
| M stage (%) | | | |
| M1a | 27.3 | 23.5 | 25.0 |
| M1b | 27.3 | 35.3 | 32.1 |
| M1c | 18.2 | 29.4 | 25.0 |
| M1d | 27.3 | 11.8 | 17.8 |
| LDH level (U/L) | 423.9 ± 150.0 | 401.2 ± 124.1 | 418.7 ± 134.7 |
| Previous therapy (%) | 0 | 0 | 0 |
| Tumor burden (cm, range) | 10.5 (2.1–43.6) | 10.5 (1.5–37.8) | 10.4 (1.5–43.6) |

**Blood collection**. All human blood samples (up to 30 mL) were obtained via informed consent through the Azienda Ospedaliero Universitaria di Modena and Reggio Emilia. Approval of study protocols was obtained by the ethical committee (Prot AOU 0005400/18). Blood was obtained before therapy (hereafter indicated as T0), after the first and the second cycle of therapy (hereafter indicated as T1 and T2, respectively), just before the drug infusion. Peripheral blood mononuclear cells (PBMC) were isolated according to standard procedures and stored in liquid nitrogen until use[40]. The whole experimental procedure is represented in Supplementary Fig. 1.

**Polychromatic flow cytometry**. A 30-parameter/28-color flow cytometry panel was optimized to broadly characterize T-cell differentiation and activation along with markers that are target or are involved in immunotherapy response (CD3, CD4, CD8, CD45RA, CD197, CD28, CD27, CD127, CD95, CD98, CD71, CD25, HLA-DR, CD38, CD39, CXCR6, CCR4, Ki67, T-bet, granulysin, PD-1, BTLA, CD244, and ICOS). Moreover, the panel was optimized to identify the expression of PD-1 in T cells isolated from patients treated with anti-PD-1 (either nivolumab or pembrolizumab) as anti-IgG4 was used to recognize the anti-PD-1 bound to PD-1[27].

Briefly, cryopreserved samples were thawed in R10 medium, i.e., RPMI supplemented with 10% fetal bovine serum (FBS), 100 U/mL penicillin, 100 µg/mL streptavidin, 2 mM L-glutamine, 20 mM HEPES (ThermoFisher, Eugene, OR) and 20 µg/mL DNase I from bovine pancreas (Sigma-Aldrich, St. Louis, MO). After washing with phosphate buffer saline (PBS), cells were stained immediately with the Zombie Aqua Fixable Viability kit (BioLegend, San Diego, CA) for 15 min at room temperature. Then, cells were washed and stained with the combination of monoclonal antibodies (mAbs) purchased from either Becton Dickinson Biosciences (BD, San José, CA), BioLegend, or eBioscience/ThermoFisher (Eugene, OR), as listed in Supplementary Table 1, that reports also the fluorochromes bound to the different monoclonal mAbs, that had been previously titrated to define the optimal concentration. Chemokine receptors were stained for 20 min at 37 °C, for 20 min at room temperature. Intracellular detection of Ki67, granulysin and T-bet was performed following fixation of cells with the FoxP3 transcription factor staining buffer set (eBioscience/ThermoFisher) according to manufacturer's instructions and by incubating with specific mAbs for 30 min at 4 °C. Samples were acquired on a FACS Symphony A5 flow cytometer (BD Biosciences) equipped with five lasers (UV, 350 nm; violet, 405 nm; blue, 488; yellow/green, 561 nm; red, 640 nm) and capable to detect 30 parameters. Flow cytometry data were compensated in FlowJo by using single stained controls (BD Compbeads incubated with fluorochrome-conjugated antibodies)[41]. Gating strategy is shown in Supplementary Fig. 1.

A 18-parameter/16-color flow cytometry panel was then optimized to broadly investigate mucosal invariant associated T (MAIT) cell phenotype, including CD3, CD8, TCR Vα7.2, CD161, CD45RO, CD197, CD28, CD27, CD127, CD95, CD25, HLA-DR, CD38, CXCR4, Ki67, granulysin, CD69. Briefly, cryopreserved samples were thawed and stained immediately with PromoFluor-840, viability probe (PromoCell - PromoKine) for 20 min at room temperature. Then, cells were washed and stained with the combination of mAbs purchased from either BD Biosciences, BioLegend, or eBioscience, as listed in Supplementary Table 2. mAbs were previously titrated to define the optimal concentration. Chemokine receptors were stained for 20 min at 37 °C, whereas all the other markers were stained for 20 min at room temperature. Intracellular detection of Ki67 and granulysin was performed following fixation of cells with the FoxP3/ transcription factor staining buffer set (eBioscience, ThermoFisher) according to manufacturer's instructions and by incubating with specific mAbs for 30 min at 4 °C. Samples were acquired on a Cytoflex LX flow cytometer (Beckman Coulter, Hialeah, FL) equipped with six lasers (UV, 355 nm; violet, 405 nm; blue, 488; yellow/green, 561 nm; red, 638 nm; IR, 808 nm) and capable to detect 21 parameters. Flow cytometry data were compensated in FlowJo by using single stained controls, as above[41]. Gating strategy is shown in Supplementary Fig. 6.

In parallel, thawed PBMC were rested for 4 h at 37 °C and then in vitro stimulated with anti-CD3/CD28 (1 µg/mL) (Miltenyi, Bergisch Gladbach, Germany) and suboptimal concentration of IL-12 (2 ng/mL) (Miltenyi) and IL-18 (50 ng/mL) (R&D System, Minneapolis, MN) and a combination of those[25]. A 11 parameter/10-color flow cytometer panel was optimized to identify MAIT cells producing Granzyme (GRZM)-A, GRZM-B, TNF and IFN-γ that were detected after 16 h of incubation (Supplementary Table 3). For the quantification of intracellular cytokines, cells were fixed with BD Cytofix/Cytoperm Fixation/ Permeabilization Solution kit (BD Biosciences) according to the manufacturer's instructions. Samples were acquired on an Attune NxT acoustic flow cytometer (ThermoFisher) equipped with four lasers (violet, 405 nm; blue, 488; yellow/green, 561 nm; red, 640 nm) and capable to detect 14 parameters. Flow cytometry data were compensated in FlowJo by using single stained controls as above. Gating strategy is shown in Supplementary Fig. 1.

**High-dimensional flow cytometry data analysis**. Flow Cytometry Standard (FCS) 3.0 files were analyzed using FlowJo software version 9.6. Aggregates and dead cells were removed from the analyses and identify CD3+CD8+ T cells were gated. 10,000 CD8+ T cells per sample were exported and biexponentially transformed in FlowJo version 10. Further analyses were performed by a custom-made

script that makes use of Bioconductor libraries and R statistical packages[4]. Data were analyzed using the Phenograph algorithm coded in the Cytofkit package (version 1.6.5;[42]) in R (version 3.3.3). Parameter K was set at 60. Phenograph clusters were visualized using tSNE. Clusters representing <0.5% were not analyzed in subsequent analysis. New FCS files (one for each cluster), originated from Phenograph analyses, were imported and analyzed in FlowJo to determine the frequency of positive cells for each marker and the corresponding median fluorescence intensity (MFI). These values were multiplied to derive the integrated MFI (iMFI, rescaled to values from 0 to 100). gplots R package was used to generate the heat map, showing the iMFI of each marker per cluster[4,43].

**Cell Sorting and single-cell RNA-sequencing (scRNA-seq) library preparation**. Cryopreserved samples were thawed in R10 supplemented with 20 µg/mL DNase I from bovine pancreas (Sigma-Aldrich). After washing with phosphate buffer saline (PBS), cells were stained with the Red Live Dead Fixable Viability kit (Thermo-Fisher) for 15 min at room temperature. PBMC were washed with PBS and stained with mAb anti-CD3-PE and -CD8-FITC. Viable CD3+CD8+ T cells were sorted by using eS3 sorter (Bio-Rad Laboratories, Hercules, CA) equipped with two lasers (blue, 488; yellow/green, 561 nm; all tuned at 100 mW). Cell sorting was performed with 96-99% purity. Sorted CD3+CD8+ T cells were immediately loaded on ddSEQ single-cell isolator (Bio-Rad Laboratories) to isolate single cells and barcode single cells. sc-RNA-seq libraries were prepared by using the Illumina Bio-Rad SureCell WTA 3' Library Prep Kit (Illumina, San Diego, CA, manufactured for Bio-Rad) following manufacturer's instructions. Briefly, after barcoding, RNA was reverse transcribed and cleaned up. Then, second strand cDNA was synthesized and tagmented. Tagmented DNA was amplified and final indexed libraries were quantified by using the high sensitivity DNA kit (Agilent, Santa Clara, CA) on a bioanalyzer (Agilent). Sequenced libraries were loaded on an Illumina NextSeq 550.

**scRNA-seq analysis**. A total of 74,404 single cells were obtained from 20 patients (9 NR and 11 R at T0, T1, T2 therapy cycles) after SureCell RNA Single-Cell (v 1.1.0) pre-processing and UMI quantification. Downstream analysis was performed in R using Seurat v3.0[44]. A Seurat object containing all 74,404 was created, then the dataset was split in three subsets T0, T1, T2. For each subset, unwanted cells were filtered out. In particular, cells that express >10% of mitochondrial genes, cells having <130 or >1500 detected genes, cells having >2500 UMI count and cells in which ribosomal protein-coding genes represented >65% of gene content, were removed. Next, feature expression-measured were normalized by dividing them by the total expression in each cell and multiplying by a factor of 10,000 (Log-Normalize). To promote the identification of common cell types and enable comparative analyses, the three datasets were integrated using IntegrateData function yielding an expression matrix of 56,142 cells by 17,745 genes[45].

Principal components were selected using the jackstraw and Elbow methods. The dimensional reduction was performed using Uniform Manifold Approximation and Projection (UMAP) on the previously selected principal components. Unsupervised clustering was performed by finding the K-nearest neighbors (KNN) and then, to group the cells, a modularity optimization-based algorithm was applied. A cluster of 231 cells featuring genes related to the myeloid lineage (expressing LYZ) and 4210 Natural Killer cells (expressing TYROBP) was excluded from the analysis for a final matrix of 51,701 cells. The resolution was selected using clustree package[46]. Differentially expressed genes were identified using the FindAllMarkers function, and the top 15 genes for each cluster were visualized in a heatmap. Differential expression analysis was performed between each cluster and all other cells using a Wilcoxon rank-sum test. Genes were selected to be significant as logFC > 0.3 and adjusted p-value < 0.05. Cells from a single cluster were selected and re-clustered to identify the presence of subpopulation. Comparative analyses across conditions inside of each cluster was performed using FindMarkers, genes were considered as significant with logFC > 0.3 and adjusted p-value < 0.05. Furthermore, a random subset was performed on all 51,701 cells selecting 4000 cells and then a trajectory analysis was performed using Monocle v2[47].

**cTP-net analysis**. The surface protein imputation was performed using a pre-trained deep neural network (cTP-net) trained on PBMC processed using multi-omics approach (CITE-seq and REAP-seq)[21]. cTP-net predict the following list of surface proteins: CD3, CD4, CD45RA, CD45RO, CD16, CD14, CD11c, CD19, CD8, CD34, CD56, CD57, CD2, CD11a, CD123, CD127-IL7Ra, CD161, CD27, CD278-ICOS, CD28, CD38, CD69, CD79b and HLA-DR. The imputation of surface proteins on our dataset was performed using integrated and normalized data.

**In silico analysis**. The scRNAseq data were retrieved from the Gene Expression Omnibus (GSE) 148190. The analysis was restricted to K383, K409 and K411 samples containing blood (B), lymph nodes metastasis (LN) or Tumor (T) data. The dataset used were GSM4455931, GSM4455932, GSM4455933, GSM4455935, GSM4455937 and GSM4455938. Data from each dataset were cleaned selecting the cells expressing <10% of mitochondrial genes, read counts of at least 200 genes and <3000 genes. Then all dataset were integrated and normalized yielding a total of 26,757 (11,614 of B, 12,915 of LN and 2170 of T).

We performed clustering and dimensional reduction using UMAP (see methods) finding ten clusters at the resolution of 0.3. Signature of each cluster was obtained by using "FindConservedMarkers" function coded in the Seurat R package. MAIT signature was confirmed by using GeneOverlap[48].

CD8[+] T cells derived from 48 human melanoma samples (accession number GSE120575)[10] were re-analyzed using python-based Scanpy (version 1.6.0)[50]. The top 1200 variable genes were selected based on normalized dispersion. The log-transformed, TPM-normalized output matrix was used for dimensionality reduction and clustering analysis. Instead, all genes were used for testing differential expression. A single-cell neighborhood graph was computed on the 50 first principal components that sufficiently explain the variation in the data using ten nearest neighbors. Resulting data were visualized using UMAP. For clustering, louvain clustering[51] at 0.6 resolution was used and cell types were annotated based on the expression of known marker genes. Characteristic gene signatures were identified by testing for differential expression of a subgroup against all other cells using a Wilcoxon rank-sum. $p$-values were corrected with "Benjamini–Hochberg" method implemented in the *tl.rank_genes_groups* function. Overlap score between signatures was computed with *sc.tl.marker_gene_overlap* function using method = "overlap_count" and normalize = "reference". Results from the analysis is reported in Source Data File and Supplementary Fig. 1.

**Statistical analysis**. Statistical analyses were performed using GraphPad Prism version 6 (GraphPad Software Inc., La Jolla, USA), unless specified otherwise. Significance of differences for the frequency of single Phenograph clusters was determined using two-way ANOVA with Bonferroni post-hoc test. To compare distributions of manually gated subsets significance was determined by paired Wilcoxon $t$-test, unless otherwise specified in the figure legends. Simplified Presentation of Incredibly Complex Evaluation (SPICE) software (version 6, kindly provided by Dr. Mario Roederer, Vaccine Research Center, NIAID, NIH, Bethesda, MD, USA) was used to analyze flow cytometry data on T-cell polyfunctionality[49]. Comparison of the curves of response to therapy was performed by Log-Rank (Mantel-Cox) test and $p$-value was considered significant <0.05. Finally, dividing patients by gender did not result in any statistical difference.

**Reporting summary**. Further information on research design is available in the Nature Research Reporting Summary linked to this article.

## Data availability

Original.fcs files concerning cytofluorimetric analysis (Figs. 1 and 4) are deposited at the flowrepository.org in the following folders: FR-FCM-Z36A; for MAIT cells: FR-FCM-Z36B The scRNA-seq data have been deposited in GEO under the accession code GSE166181. Source data are provided with this paper.

## Code availability

https://github.com/DomenicoSkyWalker89/CD8-T-lymphocytes-MAIT

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

## Acknowledgements

This paper was strongly supported by a generous, unrestricted grant from Bio-Rad Laboratories (Hercules, CA, USA). This work was also supported by Fondazione Cassa di Risparmio di Modena to G.P. (2017); by Fondo di Ateneo per la Ricerca (FAR) 2017 to A.C.; by unrestricted grants to A.C. from: Sanfelice 1873 Banca Popolare (San Felice sul Panaro, Modena, Italy), Rotary CLUB Distretto 2072 (Modena, Modena L.A. Muratori, Carpi, Sassuolo, Castelvetro di Modena, Italy). E.L. was supported by grants from the Associazione Italiana per la Ricerca sul Cancro (IG 20676 and 5×1000 UniCanVax 22757). S.P. and J.B. were supported by Fellowships from the Fondazione Italiana per la Ricerca sul Cancro-Associazione Italiana per la Ricerca sul Cancro (FIRC-AIRC). We thank Paola Castagnoli (Toscana Life Sciences), Paola Paglia (ThermoFisher), Leonardo Beretta (Beckman Coulter), Federica De Paoli and Federico Colombo (Humanitas Clinical and Research Center, Rozzano, Milan, Italy) for the precious technical advices and continuous support. E.L., S.D.B., and L.G. have been or are Marylou Ingram Scholar of the International Society for Advancement of Cytometry (ISAC) for the period 2012–2016, 2016–2020, and 2020–2024, respectively. Finally, we are grateful to all the patients who donated blood for this study.

## Author contributions

S.D.B., L.G., D.L.T. performed experiments and data analyses; E.M., M.F., S.B., M.P. performed data analyses; R.D., G.P., R.S. enrolled the patients; B.W., K.K., J.B., S.P., C.R., C.L. helped in setting up the methodology; S.D.B., L.G., E.L., and A.C. designed the study; all authors discussed the data; S.D.B., L.G., M.D., E.L. and A.C. wrote the paper.

## Competing interests

B.W. and K.K. are working for the company Bio-Rad Laboratories, which supported this study with a fully unrestricted grant, and had no role in the selection of the patients and in deciding the assays that were performed. The other authors declare no competing interests.
