## [Peer Review File · Nature Communications]

REVIEWER COMMENTS

Reviewer #1 (Melanoma immunity, T cell activation) (Remarks to the Author):

Using multiparametric flow cytometry and single cell RNA sequencing of PBMC samples, the authors performed immune system profiling of 2 groups of patients with metastatic melanoma responding (R) and non-responding (NR) to the standard-of-care nivolumab or pembrolizumab treatment. They showed a significant higher proportion of proliferating effector memory cells population before and after the 2 course of immune checkpoint inhibitor (ICI) treatment in the patients responsive to the drug. More detailed analysis shows that while activated CD8 T cell compartment was significantly increased after the second course of ICI, an activated/functional MAIT cell population was present as a higher proportion in the blood of responder patients. The increase in MAIT cells was seen in responders before treatment. The authors suggest the percentage of activated MAIT population as a predictive biomarker for ICI response.

Suggested corrections and comments:

- Line 621: Figure1B legend indicates "Individual values of cells present in each cluster". I would suggest "percentage of cell within each cluster for each individual"
- Figure1B: The Y axis title shows "% of cluster" only for T2. The title is missing for T0 and T1. Nomenclature on figure and in figure legend should be the same (see above).
- Line 628: Figure 2A legend should indicate that UMAP was done on the CD8 population.
- Figure 2D: The difference in % of MAIT cells at T2 seems to be statistically different too. Nomenclature on figure and in figure legend should be homogenized ("percentage" vs "proportion").
- Figure 4B: The single cell data are from very different tissue samples and pooled together for the analysis (same UMAP). As quality control of the clustering, it would be informative to see the organ distribution over the UMAP and absolute numbers analyzed by organ. It would be also informative to know which markers the authors used to quantitate the percentage of MAIT cells in the right histogram.
- Figure 4C: The figure legend should indicate how many patients sample and which timepoints were used for the assay. This is valuable for the interpretation of the data. On the histogram, it seems that the error bars are not correctly represented. There is an extra "#" above the histogram. In the legend (Line 666), the authors mention IL-2 production in the figure legend but it is not represented in the figure 4C. and GZMB is not mentioned in the figure legend.
- Figure 4D: The figure legend does not indicate which time point is shown on the dot plot. It seems that difference between R and NR in % of GZMB and IFNG double positive and IFNG single positive cells are as well as significant than GZMB+ cells. This should be described in the main text.
- Figure 5: The figure legend does not indicate number of each patient in each group (> or < 1.7% of MAIT cells).

Overall, Flow data and single cell RNAseq analysis were technically robust. Biological differences claimed by the authors are supported by statistical significance.

With checkpoint inhibitors becoming standard of care for melanoma treatment, a molecular and

cellular knowledge of the immune system profiles in response to treatment is critical to the field. The authors here bring to light a significant difference on the MAIT cell population.

Although the role of MAIT cells is controversial, the data presented here are complete and robust. These findings could pave the way to development of new ICI targeting strategies. Moreover, with the emergence of autoimmune adverse events reported in ICI treated patients, these data are bringing new insight in the establishment of the ICI induced autoimmune adverse events reported in ICI treated patients, these data are bringing new insight in the establishment of the ICI induced autoimmune reactions.

Reviewer #2 (T cell biology, systems immunology) (Remarks to the Author):

1. Age and sex are known to affect the frequency of different immune cell type. It may be useful to show none of the clinical parameters in table 1 is affecting response, especially since sex ratio is very different between R and NR.
2. All two sided Mann-Whitney nonparametric tests done in figures 1-4 are not explicitly performed with correction for multiple comparisons. Please specify correction used, and if no correction applied, make sure to apply one.
3. L110 - There is no longitudinal analysis – R and NR are compared in each time point. It may be interesting to see changes in frequencies of cell type between points.
4. L193 – It is unclear if this analysis was performed on an independent cohort of patients or not. If it is an independent cohort, it should be clarified, and the cohort should be described. If it is the same cohort used to identify MAIT levels as different between R and NR, it is circular, trivial, and might even be misleading. In that case, possibly patients with different levels of response can be compared.
5. L212 – where was a differential dynamics of T cells shown?
6. L467 – T1 was collected after the first and second cycle of therapy, but how long after?
7. L557 – filtering of cells is standard, but the wording is confusing. Also, specific values used to filter the cells should be stated, so that the analysis can be repeated by others.
8. L592 – as far as I can see, no Gene set Enrichment Analysis (GSEA) was done, and this method section certainly does not describe how GSEA analysis was done.
9. Figure 1 – significance level not specified. In B, the meaning of the Y axis is unclear. For each individual, percentage of his/her cells that are in this cluster? Out of all patient cells? Patient cells in the shown time point?
10. Figure 2 C&D right – how were the differentially expressed genes identified, and how are they marked in these figures? How were the genes whose names are given selected? Should be stated in figure legend. There are genes that seem to have as big a fold change to the other direction and their names are not given. It seems there is no table with all genes that are significant in each of these comparisons.
11. L636 – no individual values are presented. May be more convincing to present individual values, as in figure 1B.

REVIEWER COMMENTS

Reviewer #1 (Melanoma immunity, T cell activation) (Remarks to the Author):

Using multiparametric flow cytometry and single cell RNA sequencing of PBMC samples, the authors performed immune system profiling of 2 groups of patients with metastatic melanoma responding (R) and non-responding (NR) to the standard-of-care nivolumab or pembrolizumab treatment. They showed a significant higher proportion of proliferating effector memory cells population before and after the 2 course of immune checkpoint inhibitor (ICI) treatment in the patients responsive to the drug. More detailed analysis shows that while activated CD8 T cell compartment was significantly increased after the second course of ICI, an activated/functional MAIT cell population was present as a higher proportion in the blood of responder patients. The increase in MAIT cells was seen in responders before treatment. The authors suggest the percentage of activated MAIT population as a predictive biomarker for ICI response.

Suggested corrections and comments:

• **Line 621: Figure1B legend indicates “Individual values of cells present in each cluster”. I would suggest “percentage of cell within each cluster for each individual”**

The legend has been changed according to the reviewer’s suggestion.

• **Figure1B: The Y axis title shows “% of cluster” only for T2. The title is missing for T0 and T1. Nomenclature on figure and in figure legend should be the same (see above).**

“proportion of cluster” has been added also for T0 and T1, and title has been added.

In figure 1A and 1B the name of the clusters is reported in the same way (C1, C2...).

• **Line 628: Figure 2A legend should indicate that UMAP was done on the CD8 population.**

Now this is reported as “UMAP plot depicting CD8 T cell heterogeneity”.

• **Figure 2D: The difference in % of MAIT cells at T2 seems to be statistically different too.**

Nomenclature on figure and in figure legend should be homogenized (“percentage” vs “proportion”).

Nomenclature has been homogenized and “proportion” has been used. The difference indicated by the referee, even if evident, was not statistically significant, and for this reason it was not indicated.

· **Figure 4B: The single cell data are from very different tissue samples and pooled together for the analysis (same UMAP). As quality control of the clustering, it would be informative to see the organ distribution over the UMAP and absolute numbers analyzed by organ. It would be also informative to know which markers the authors used to quantitate the percentage of MAIT cells in the right histogram.**

We thank the referee for this suggestion. The organ distribution over the UMAP is now represented in figure 4B (right panel) and it is described in the figure legend. Here below, we report the absolute numbers analyzed by organ. MAIT cells were identified by the gene signature reported in “supplementary file gene signature”.

SAMPLE	MAIT count	TOTAL count	percentage
K411_LN	95	3870	2,5
K411_B	494	6288	7,9
K409_LN	212	6144	3,5
K409_B	206	5326	3,9
K409_T	76	2170	3,5
K383_LN	95	2953	3,2

· **Figure 4C: The figure legend should indicate how many patients sample and which timepoints were used for the assay. This is valuable for the interpretation of the data. On the histogram, it seems that the error bars are not correctly represented. There is an extra “#” above the histogram. In the legend (Line 666), the authors mention IL-2 production in the figure legend but it is not represented in the figure 4C. and GZMB is not mentioned in the figure legend.**

We agree with the reviewer and we apologize for missing this information. Numbers of patients are now indicated in each figure legend.

We have corrected the figure as suggested:

- timing is indicated

- “#” has been replaced with “*” as the difference is statically significant (p=0.041)
- IL-2 has been removed as it was a mistake.

Those shown in the figure there are not error bars, but the upper whiskers lines (whisker range, SEM). The reason is that error bars represented by the graphic used by SPICE are confusing, thus we would prefer to use this representation.

· Figure 4D: The figure legend does not indicate which time point is shown on the dot plot. It seems that difference between R and NR in % of GZMB and INFG double positive and IFNG single positive cells are as well as significant than GZMB+ cells. This should be described in the main text.

We agree with the reviewer and we apologize for the lack of information. The time point is now reported in the figure. The proportion of MAIT cells producing IFN- γ and GZMB is higher in R if compared to NR, and this has been added in figure 4C. The proportion of MAIT cells producing IFN- γ was similar between R and NR.

· Figure 5: The figure legend does not indicate number of each patient in each group (> or < 1.7% of MAIT cells).

According to the reviewer, the number of patients for each group have been shown.

Overall, Flow data and single cell RNAseq analysis were technically robust. Biological differences claimed by the authors are supported by statistical significance.

With checkpoint inhibitors becoming standard of care for melanoma treatment, a molecular and cellular knowledge of the immune system profiles in response to treatment is critical to the field. The authors here bring to light a significant difference on the MAIT cell population.

Although the role of MAIT cells is controversial, the data presented here are complete and robust. These findings could pave the way to development of new ICI targeting strategies.

Moreover, with the emergence of autoimmune adverse events reported in ICI treated patients, these data are bringing new insight in the establishment of the ICI induced autoimmune adverse events reported in ICI treated patients, these data are bringing new insight in the establishment of the ICI induced autoimmune reactions.

We thank the referee for these words of appreciation of our work.

Reviewer #2 (T cell biology, systems immunology) (Remarks to the Author):

1. Age and sex are known to affect the frequency of different immune cell type. It may be useful to show none of the clinical parameters in table 1 is affecting response, especially since sex ratio is very different between R and NR.

We thank the reviewer for this comment that give us the possibility to clarify that we performed the statistical analysis dividing patients by gender and by response to therapy. We saw however that gender does not affect the response to therapy, and added a short sentence at the end of the Statistical analysis section: “Finally, dividing patients by gender did not result in any statistical difference”.

2. All two-sided Mann-Whitney nonparametric tests done in figures 1-4 are not explicitly performed with correction for multiple comparisons. Please specify correction used, and if no correction applied, make sure to apply one.

We thank the reviewer for this comment. We used Bonferroni's multiple comparisons test after two-sided Mann-Whitney nonparametric test. We add this to figure legends.

3. L110 - There is no longitudinal analysis – R and NR are compared in each time point. It may be interesting to see changes in frequencies of cell type between points.

We clarified that longitudinal analysis was performed, but it did not revealed statistical differences and we change the text accordingly:

“Longitudinal analysis did not identify obvious differences in the dynamics of these immune populations between responders and non-responders to anti-PD-1 therapy (data not shown). Cross-sectional analysis identified C16, highly proliferating Ki-67+ CD71+ effector cells equipped for cytotoxicity (GNLY+), whose relative frequency was higher in responder before starting therapy ($p < 0.001$) (Figure 1B). This difference remained stable also after treatment ($p < 0.01$) (Figure 1B).”

4. L193 – It is unclear if this analysis was performed on an independent cohort of patients or not. If it is an independent cohort, it should be clarified, and the cohort should be described. If it is

the same cohort used to identify MAIT levels as different between R and NR, it is circular, trivial, and might even be misleading. In that case, possibly patients with different levels of response can be compared.

The total cohort of patient studied is composed by 11 NR and 17 R. Unfortunately, due to lack of a sufficient number of thawed PBMC, we could not perform all experiments that we had thought. Of this cohort (28 patients), we used biological material from 8 NR and 9 R for CD8 immunophenotype, 8 NR and 11 R for sc-RNAseq. Flow cytometry experiments for MAIT confirmation were performed on additional patients recruited in the last period of the study, and experiments were performed on 4 NR and 6 R. We tried to perform additional analysis by dividing different level of response to therapy, but due to low number of patients, obviously statistical analysis was not as robust as we had wished.

According to the reviewer's suggestion, we focused our attention on a published dataset of sc-RNAseq on melanoma patients treated with ICI, and then we could reinforce the analysis we had performed in our cohort of patients. In fact, the importance of MAIT cells as biomarker able to predict the response to therapy has been further underlined by this additional analysis performed on a previously published dataset of CD8+ T cells derived from 48 human melanoma samples in which MAIT signature was originally not pointed out. As shown in supplementary figure 6 (bottom panel), we identified a signature of MAIT cells characterized by the expression of their characteristic gene *KLRB*. We found indeed that after therapy MAIT cells increased in the metastatic lesions of R and not in NR.

5. L212 – where was a differential dynamics of T cells shown?

We thank the reviewer for this comment. We apologize for the misleading sentence, that we have changed as below:

“Following introduction of ICI, R show different proportions of T cells compared to NR, involving the expansion of activated effector memory cells showing features of immune activation, proliferation and effector differentiation, as previously reported by other groups.”

6. L467 – T1 was collected after the first and second cycle of therapy, but how long after?

We thank the reviewer for this comment, and we clarified that the blood was collected just before the drug infusion.

“Blood was obtained before therapy (hereafter indicated as T0), after the first and the second cycle of therapy (hereafter indicated as T1 and T2 respectively), just before the drug infusion.”

7. L557 – filtering of cells is standard, but the wording is confusing. Also, specific values used to filter the cells should be stated, so that the analysis can be repeated by others.

We apologize if the text was not clear. In the “Methods” section we have now specified how cells were filtered, as follows:

sc-RNAseq analyses

A total of 74,404 single-cell were obtained from 20 samples (9 NR and 11 R at T0, T1, T2 therapy cycles) after SureCell RNA Single-Cell (v 1.1.0) pre-processing and UMI quantification. Downstream analysis was performed in R using Seurat v3.0⁴⁴. A Seurat object containing all 74,404 was created, then the dataset was split in three subsets T0, T1, T2. For each subset, unwanted cells were filtered out. In particular, cells that express more than 10% of mitochondrial genes, cells having less than 130 or more than 1,500 detected genes, cells having more than 2500 UMI count and cells in which ribosomal protein-coding genes represented more than 65% of gene content, were removed. Next, feature expression-measured were normalized by dividing them by the total expression in each cell and multiplying by a factor of 10,000 (LogNormalize). To promote the identification of common cell types and enable comparative analyses, the three datasets were integrated using IntegrateData function yielding an expression matrix of 56,142 cells by 17,745 genes⁴⁵.

Principal components were selected using the jackstraw and Elbow methods. The dimensional reduction was performed using Uniform Manifold Approximation and Projection (UMAP) on the previously selected principal components. Unsupervised clustering was performed by finding the K-nearest neighbors (KNN) and then, to group the cells, a modularity optimization-based algorithm was applied. A cluster of 231 cells featuring genes related to the myeloid lineage (expressing *LYZ*) and 4,210 Natural Killer cells (expressing *TYROBP*) was excluded from the analysis for a final matrix of 51,701 cells. The resolution was selected using clustree package⁴⁶. Differentially expressed genes were identified using the FindAllMarkers function, and the top 15 genes for each cluster were visualized in a heatmap. Differential expression analysis was performed between each cluster and all other cells using a Wilcoxon rank-sum test. Genes were selected to be significant as $\log_{2}FC > 0.3$ and adjusted p value < 0.05 . Cells from a single cluster were selected and re-clustered to

identify the presence of subpopulation. Comparative analyses across conditions inside of each cluster was performed using FindMarkers, genes were considered as significant with $\log_{2}FC > 0.3$ and adjusted p value < 0.05 . Furthermore, a random subset was performed on all 51,701 cells selecting 4,000 cells and then a trajectory analysis was performed using Monocle v2⁴⁷.

8. L592 – as far as I can see, no Gene set Enrichment Analysis (GSEA) was done, and this method section certainly does not describe how GSEA analysis was done.

We thank the reviewer for this note, and apologize for the mistake. GSEA was not included in the paper neither the methods, and we have replaced GSEA with “*In silico* analysis”.

9. Figure 1 – significance level not specified. In B, the meaning of the Y axis is unclear. For each individual, percentage of his/her cells that are in this cluster? Out of all patient cells? Patient cells in the shown time point?

Significance level is now specified, as well as the meaning of Y axis.

The figure legend has been modified as follows:

“A) Heatmap showing the iMFI of specific markers in discrete Phenograph clusters. Ballons indicate the median frequency of each cluster amongst responders and non-responders. Statistical analysis by two-sided Mann Whitney nonparametric test, Bonferroni's multiple comparison test, * $p < 0.05$. B) Proportion of cell within each cluster for each individual. Data represent individual values, mean (centre bar) \pm SEM (upper and lower bars). Statistical analysis by two-sided Mann–Whitney nonparametric test, Bonferroni's multiple comparisons test; * $p < 0.05$; *** $p < 0.001$. T0= before therapy, T1=after 1 cycle of therapy, T2=after two cycles of therapy.”

10. Figure 2 C&D right – how were the differentially expressed genes identified, and how are they marked in these figures? How were the genes whose names are given selected? Should be stated in figure legend. There are genes that seem to have as big a fold change to the other direction and their names are not given. It seems there is no table with all genes that are significant in each of these comparisons.

We thank the referee for this comment that allowed us to improve significantly Figure 2 and to add all the tables with the differential analyses between time points, responders and not

responders for each cluster. In figure 2 C and D, only the differentially expressed genes are reported. P values are indicated in the new supplementary tables marked as source data file.

11. L636 – no individual values are presented. May be more convincing to present individual values, as in figure 1B.

We thank the reviewer for the suggestion, and now individual values are reported as in Figure 1.

REVIEWERS' COMMENTS

Reviewer #1 (Remarks to the Author):

The authors have addressed all the concerns and made the changes that improved comprehension of the data. The two main concerns were that the linkage of %of MAIT cells with the % of patient response was done on a very low number of patients leading to a conclusion to use MAIT population as a biomarker for ICI response and the lack of evidence of a direct role of MAIT cells in the antitumoral action of ICI treatment. The authors have addressed those limitations in the discussion. Nevertheless, the increase of the MAIT population in the responder's group is robust. Therefore, I have no further comments or objections to the data presented here.

Reviewer #2 (Remarks to the Author):

All comments addressed satisfactorily.